# Low-Frequency Noise-Based Mechanism Analysis of Endurance Degradation in Al/αTiO_x_/Al Resistive Random Access Memory Devices

**DOI:** 10.3390/ma16062317

**Published:** 2023-03-14

**Authors:** Jung-Kyu Lee, Juyeong Pyo, Sungjun Kim

**Affiliations:** Division of Electronics and Electrical Engineering, Dongguk University, Seoul 04620, Republic of Korea

**Keywords:** resistive switching, RRAM, Al/αTiO_x_/Al, degradation, low-frequency noise, noise power spectral density, 1/*f* noise

## Abstract

In this work, we analyze a resistive switching random access memory (RRAM) device with the metal–insulator–metal structure of Al/αTiO_x_/Al. The transport mechanism of our RRAM device is trap-controlled space-charge limited conduction, which does not change during the endurance test. As the number of resistive switching (RS) cycles increases, the current in the low-resistance state (LRS) does not change significantly. In contrast, degradation in the high-resistance state (HRS) is noticeably evident. According to the RS cycle, the current shift fits well with the stretched-exponential equation. The normalized noise power spectral density (*S*_i_/*I*^2^) measured in the HRS is an order of magnitude higher than that in the LRS owing to the difference in the degree of trap occupancy, which is responsible for the transition of resistance states. During the consecutive RS, the *S*_i_/*I*^2^ in the HRS rapidly decreases for approximately 100 cycles and then saturates. In contrast, in the LRS, the *S*_i_/*I*^2^ does not change significantly. Here we propose a model associated with the endurance degradation of the experimental device, and the model is verified with a 1/f noise measurement.

## 1. Introduction

To apply information and communication technology such as artificial intelligence, augmented reality, virtual reality, and self-driving to our daily life, data are expected to be generated three times more than the current level [1,2,3]. In addition, as services, including telecommuting, web conferences, and streaming, have proliferated around the globe due to COVID-19, data usage will increase further. According to the statistics from international data corporations, the total amount of global data is expected to reach 175 ZB by 2025 [3]. Hence, if the demand for memory performance and capacity surges due to the explosive increase in data, the “memory wall” between the NAND flash and dynamic random access memory (DRAM), which are the current major memory technologies, will also reach an extreme [4,5,6]. Therefore, to solve the “memory wall” caused by the performance gap between the DRAM and NAND flash memory, there is an increasing need for universal memory that can implement the functions of storage memory (high density and non-volatile) and working memory (high speed) at the same time [1,7]. In recent decades, resistive switching random access memory (RRAM) has received much attention as an emerging non-volatile memory (NVM) technology because of CMOS-compatible materials, simple cell structure, good scalability (<10 nm), low switching current (~nA), and 3D integration [8,9,10]. In addition, recently, in advanced computing technologies for neuromorphic systems, RRAM has also attracted great attention as one of the promising candidates for synaptic electronics for the hardware implementation of artificial neural networks owing to its non-volatility, repeatable analog switching with good precision, and the potential for large-scale integration with the crossbar array structure [11,12,13]. Although great progress has been made recently, research on RRAM still faces some significant challenges, such as the broad distribution of the switching parameters (V_forming_, V_set_, V_reset_, R_HRS_, and R_LRS_), retention failure, and endurance degradation [14,15,16]. These issues with the device reliability stem from intrinsic variability because the resistive switching (RS) mechanism of RRAM itself is fundamentally stochastic. Consequently, a prerequisite to the successful application of RRAM is understanding the underlying physical mechanism associated with the reliability issues in RRAM. Chen et al. reported the physical mechanisms of endurance degradation in transition metal oxide-RRAM [17]. Three failure modes were experimentally identified: (1) resistive window (RW) collapse with decreased R_HRS_ (high resistance) and increased R_LRS_ (low resistance), (2) sudden stochastic RESET failure, and (3) gradual loss of the RW with a steady decrease in R_HRS_. Unfortunately, applying the mechanism associated with endurance failure behaviors is restricted to the filamentary RRAM. Therefore, a systematic study on the mechanism of endurance degradation in the interfacial RRAM is required, but little research has been conducted so far. It has already been verified in previous studies that low-frequency noise (LFN) measurements can be used as a useful tool for analyzing RRAM, such as the nature and information of the traps, the current transport, and the RS mechanism [18,19,20,21]. In this work, we investigate the physical mechanism of endurance degradation in an Al/αTiO_x_/Al interfacial RRAM device by using the LFN measurement. The RS phenomenon of the αTiO_x_ was reported by Argall in 1986 [22]. Since then, there have been many studies on the RS mechanism related to the NVM applications [23,24,25]. Recently, Jang et al. reported a learning-rate modulable and reliable TiO_x_ memristor array for robust, fast, and accurate neuromorphic computing [13]. In particular, interfacial RRAMs have the forming-free and gradual set/reset characteristics, which can reduce additional power consumption and are advantageous for the symmetry/linearity of conductivity changes, unlike filament RRAMs [26,27,28,29,30]. Therefore, our study on the degradation mechanism analysis for the optimization of Al/αTiO_x_/Al interfacial RRAM is a very valuable work. The result of the noise analysis can be direct evidence of the physical origin of the endurance failure because its characteristic is utilized to analyze the internal physics of electronics at the defect level [31,32].

## 2. Materials and Methods

Figure 1a shows the schematic structure of our experimental device. We fabricated the interfacial RRAM device with the metal–insulator–metal structure of Al/αTiO_x_/Al to investigate the physical mechanism of endurance degradation. The fabrication process is described as follows. A TiO_x_ film with a thickness of ~8 nm was deposited on a 50 nm thick Al/SiO_2_/Si substrate by a plasma-enhanced atomic layer deposition at a substrate temperature of 180 °C. Titanium tetra-iso-propoxide (TTIP) and oxygen plasma were used as the Ti and oxygen precursors, respectively. The 50 nm thick aluminum bottom and top electrodes were deposited by a thermal evaporation method, forming cross-bar-type structures using a shadow mask with a line width of 60 μm, as shown in Figure 1a. Figure 1b shows the cross-sectional transmission electron microscopy (TEM) image of the cell used in experiments.

A Keysight B1500A semiconductor parameter analyzer was employed to measure the RS characteristics of the fabricated RRAM devices. The LFN characteristics were analyzed with a low-noise current amplifier (SR570) and a signal analyzer (35670A). The voltage applied to the TiN layer was supplied using a B1500A system. The output current of the RRAM device was connected to the SR570 system, converting the current fluctuation into a voltage fluctuation. The 35670A system converted the dynamic signal from the SR570 system to a power spectral density. The noise floor of our measurement system was measured to be ~10^−24^ A^2^/Hz, which was much lower than the device noise. This guaranteed that the noise power spectral densities measured in this work were not affected by the noise floor of the measurement system.

## 3. Results

Figure 2a shows the I–V characteristics of the fabricated Al/αTiO_x_/Al RRAM device for the initial and 30 cycles. The bias sweep sequence is indicated by the arrows. When the voltage was swept from 0 V to the negative voltage direction, the device transitioned from a high-resistance state (HRS) to a low-resistance state (LRS) above the set voltage (SET process). The LRS was held up to about 2 V during the positive voltage sweep and then switched back to the HRS above the reset voltage (RESET process). As shown in Figure 2a, the device exhibited gradual set/reset characteristics that did not show abrupt current changes. A compliance current of 1 mA was applied to protect the device from dielectric breakdown. In our device, if a positive voltage was applied to the initial cell, no switching occurred, and the device permanently broke down. That is, it showed SET operation at negative voltage and RESET operation at positive voltage. These asymmetric operation characteristics of Al/αTiO_x_/Al devices can be explained by asymmetric interface formation [33]. When referring to the classification criteria of RRAM devices, our device can be classified as an interfacial RRAM device in which no conductive filament or localized path is formed within the dielectric [34,35]. The inset of Figure 2a shows the retention characteristics. Our device maintained the HRS and LRS without significant state change for 10^4^ s, which guaranteed the reliability of the LFN measurements. To clarify the current transport mechanism in both the HRS and LRS, the I–V curve at the 30th cycle is replotted on a double-logarithmic scale in Figure 2b. Referring to the linear guidelines in Figure 2b, the conduction mechanism of the device can be understood by using the space-charge limited conduction (SCLC) model [36,37]. The trap-controlled SCLC can be divided into two regions. In the low-field region, the conduction mechanism was dominated by the thermally generated free electrons in the dielectric film (Ohmic conduction, I∝V). If the applied field intensity exceeded the critical value, the density of free electrons injected from the electrode gradually exceeded the equilibrium concentration, and excess electrons accumulated in the space between the two electrodes. Consequently, the space charge started to limit the total current flow (SCLC, $I\propto V^{m},\;m>2$). Jeong et al. proposed the RS mechanism of the Al/αTiO_x_/Al RRAM device [23]. In the HRS, oxygen ions are accumulated near the top interface due to redox reactions, which increases the barrier at the interface. Although bulk αTiO_x_ has relatively high conductivity due to internal oxygen vacancies, the overall current is determined by the interface. In contrast, in the LRS, the barrier near the top interface decreases due to the drift of oxygen ions caused by the set process (positive bias). In addition, the concentration of oxygen vacancies in the bulk αTiO_x_ decreases, which makes the αTiO_x_ layer more insulating.

Switching endurance, which tells how often a memory device can switch between cell states without degradation, is one of the most important figures of merit for a memory device [38]. From this point of view, developing a technique to analyze the degradation mechanism of RRAM devices is a very good direction for RRAM optimization. Figure 3a shows the double-logarithmic plot of the I–V characteristics (HRS to LRS) for 500 DC cycles. While the current variation in the LRS was insignificant, the degradation in the HRS was noticeably clear. The change in the trend of the I–V curve, which gradually changed in one direction, was clearly different from the cycle-to-cycle variability due to the stochastic nature of the RS phenomenon [39,40,41]. To analyze the progressive RW collapse of the device, we first verified whether the current transport followed the SCLC mechanism well along the endurance cycle. Figure 3b shows the slopes of the I–V curves in Ohmic and the SCLC region according to the RS cycle. According to the RS cycle, the Ohmic region’s slope was almost constant at 1 in both resistance states. However, the slope of the SCLC region changed to 2.7, 2.57, and 2.2 at the first, 10th, and 100th periods, respectively, and then converged to 2 in the HRS. According to the SCLC theory [36,37], the slope of the I–V curve in the double logarithmic plot is expected to be 2 for a discrete trap distribution ($I\propto V^{2})$ and greater than 2 for an exponential trap distribution ($I\propto V^{m+1},\;m>1)$. So, it can be seen that the conduction mechanism does not change according to the RS cycle. In addition, the decrease in the slope with consecutive RS cycles suggests electric-field-induced charge trapping in the oxide. According to the trap-controlled SCLC model [24,25], SCLC arises if the current through the bulk solid becomes limited by the buildup of charge injected from the electrode. If the applied voltage is raised to a threshold value, at which point the number of charge carriers injected at the electrode becomes equivalent to the number of thermally generated ones in the bulk, the injected carriers are sufficient to fully fill the trap states, resulting in a rapid increment of the current. The degradation of our experimental RRAM device is closely related to this SCLC mechanism. Namely, certain traps whose energy levels are far below the conduction band (deep traps) are likely to fail to release the trapped charge carriers during the reset process. Consequently, the repetitive RS process could induce a more effective filling-up of the deep traps and prevent their effective de-trapping during the subsequent reset process. Figure 3c shows the endurance characteristic in both resistance states at the read voltage of 0.1 V. At the early stage of the RS cycle, the current increased exponentially and reached saturation with an almost linear shape. To describe the relaxation of out-of-equilibrium disordered systems that do not obey a simple exponential law, a stretched exponential (SE) function is widely used [42,43,44]. This model is defined as $f\left(x\right)=e^{-x^{\beta }}$, where x is the independent variable (here x is the cycle number), and β is the stretched exponent between 0 and 1. Figure 3d shows the results of the simple exponential (blue line) and the SE models (red line) fitting. The fitted results were more consistent with the SE function at the beginning and end of the curves as shown in the insets of Figure 3d.

Based on these results, we propose a mechanism for the endurance degradation in Al/αTiO_x_/Al interfacial RRAM devices as shown in Figure 4. When assuming a discrete trap distribution, the process of filling/emptying all traps is repeated during the RS cycle, as shown in Figure 4a,b, and no RS degradation occurs. However, when assuming exponential trap distribution, the deep traps filled with electrons during the set process do not release electrons during the reset process (Figure 4c), consequently reducing the resistance at the HRS. Our understanding of the RS degradation mechanism can be well supported by subsequent LFN measurements. This is because the change in the trap density is directly related to the noise level [45,46].

Figure 5a shows the normalized noise power spectral density (*S*_i_/*I*^2^) for several devices measured at 0.1 V in both the HRS and LRS. The *S*_i_/*I*^2^ was proportional to $1/f^{\gamma }$, with γ ~ 1 for both states in most cells, which means that the LFN characteristic in Al/αTiO_x_/Al RRAM devices also obeyed the classical 1/f noise theory in both states (see dotted line in the inset of Figure 5a) [47]. The normalized noise power measured in the HRS was an order of magnitude higher than that in the LRS. The difference in noise levels in the different resistance states can be analyzed based on our model illustrated in Figure 4. Considering that the degree of trap occupancy was responsible for the transition of resistance states, the magnitude of the *S*_i_/*I*^2^ in the LRS was reduced because all trap sites were filled by free electrons. In contrast, in the HRS, the electron flow was obstructed by the noise source, such as Columbic scattering caused by empty trap sites, which consequently increased the noise level [48]. Figure 5b shows the *S*_i_/*I*^2^ in both the HRS and LRS according to the RS cycle at the same read bias. The *S*_i_/*I*^2^ in the LRS did not change significantly with an increase in the number of cycles, while the *S*_i_/*I*^2^ in the HRS gradually decreased. Figure 5c shows the value of the *S*_i_/*I*^2^ with increasing cycle numbers at the frequencies 20, 40, and 100 Hz. In the HRS, because certain traps did not release the electrons during the reset process and thus induced a decrease in the trap density, the *S*_i_/*I*^2^ rapidly decreased for approximately 100 cycles and then saturated. In contrast, in the LRS, the *S*_i_/*I*^2^ did not change significantly. Namely, the reduction of the trap, which was the scattering center of the charge carrier [48], caused a decrease in the *S*_i_/*I*^2^. This result shows that the mechanism of the endurance degradation in Al/αTiO_x_/Al RRAM, proposed from the measured degradation characteristics and the SCLC theory, was consistent with the LFN measurement results.

## 4. Conclusions

In conclusion, we investigated the physical mechanism of the endurance degradation in Al/αTiO_x_/Al interfacial RRAM devices using the LFN measurements. The gradual RW collapse of the device was distinctly different from the abrupt RW failure of the filamentary RRAM. During the endurance test, the current transport mechanism maintained the SCLC, but the trap distribution was changed from a discrete distribution to an exponential distribution. In addition, the stretched-exponential equation was efficiently applied in fitting the current shift according to the RS cycle. The *S*_i_/*I*^2^ measured in the HRS was an order of magnitude higher than that in the LRS because of the difference in the degree of trap occupancy. From the degradation characteristic and LFN measurement, we proposed a mechanism of endurance degradation related to the electric field-induced charge trapping. The proposed model was well supported by the 1/f noise measurement according to the RS cycle, which showed that LFN measurement can be a valuable analytical tool to clarify the physical mechanism associated with the RS phenomenon of RRAM devices. Overall, this study demonstrates the usability of LFN measurements in producing direct evidence of the physical mechanism underlying the RS degradation phenomenon, which can facilitate the development of RRAM devices.

## Figures and Tables

**Figure 1 materials-16-02317-f001:**
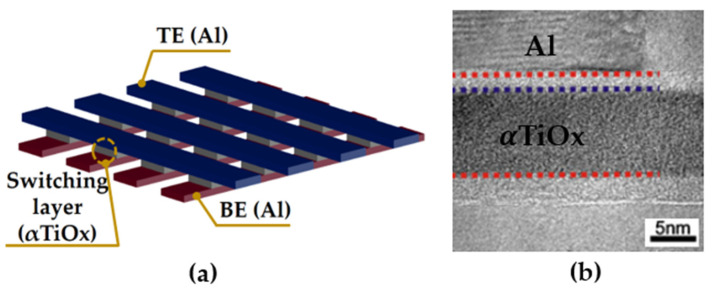
(**a**) The simple structure and (**b**) the cross-sectional TEM image of the fabricated Al/αTiO_x_/Al RRAM device.

**Figure 2 materials-16-02317-f002:**
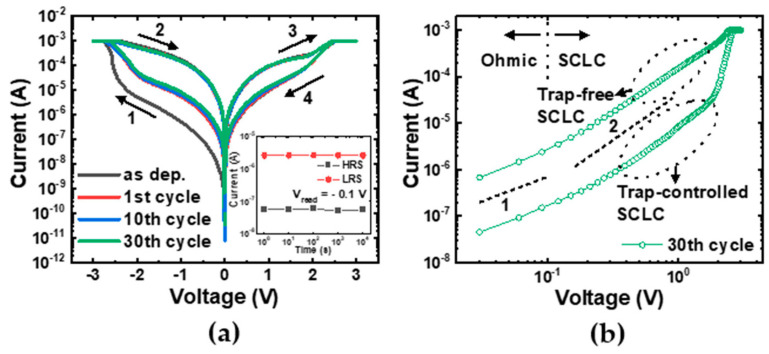
(**a**) The I–V characteristics of the fabricated Al/αTiO_x_/Al interfacial RRAM devices for the initial and 30 cycles. The arrows indicate the direction of the voltage sweep. The inset shows the retention characteristics. (**b**) The double-logarithmic plot of the I–V curve at the 30th cycle.

**Figure 3 materials-16-02317-f003:**
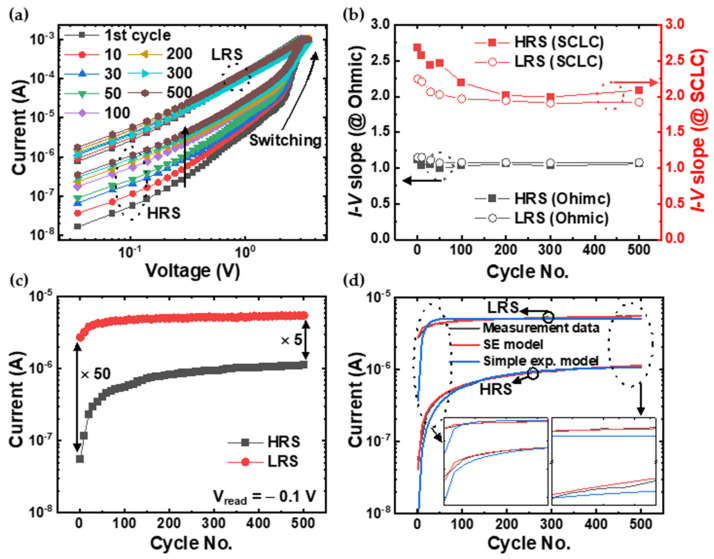
(**a**) The double-logarithmic plot of the I–V characteristics (HRS to LRS) for 500 DC cycles. (**b**) The change in slopes of the I–V curves in Ohmic (black) and the SCLC (red) region according to the RS cycle. (**c**) The endurance characteristic in both the HRS and LRS at the read voltage of 0.1 V. The initial on/off ratio of about 50 decreases to 5 after 500 DC cycles. (**d**) The fitting results for simple exponential (blue line) and SE function (red line). Insets show enlarged graphs at the beginning (left) and end (right) of the curves.

**Figure 4 materials-16-02317-f004:**
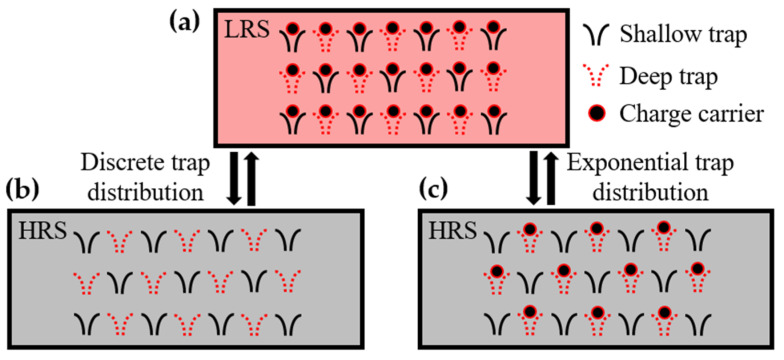
Schematic illustration explaining the physical mechanism of endurance degradation in Al/αTiO_x_/Al interfacial RRAM device. (**a**) LRS after the SET process, in which all traps are filled by the charge carriers. (**b**) HRS after the RESET process, in which all traps are emptied when assuming the discrete trap distribution. (**c**) HRS after the RESET process when assuming the exponential trap distribution. Certain traps do not release the charge carriers during the RESET process.

**Figure 5 materials-16-02317-f005:**
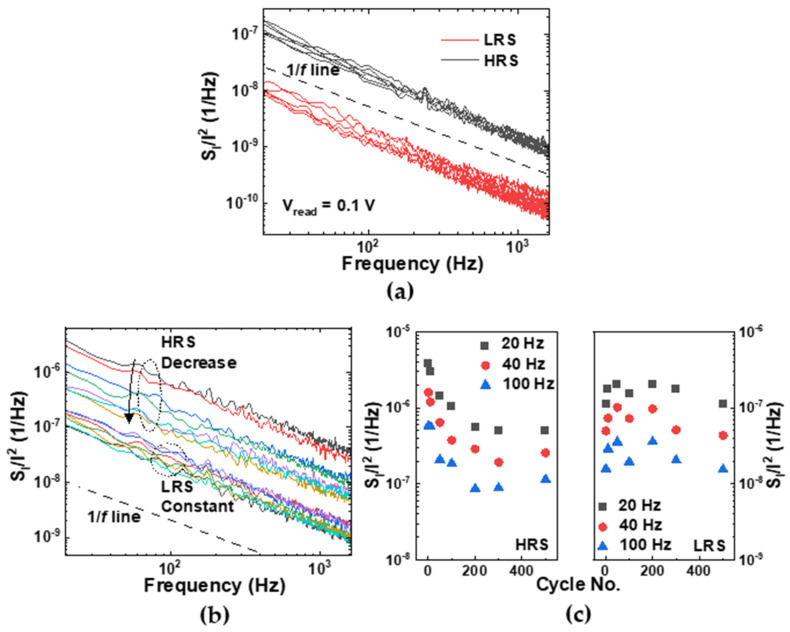
(**a**) The *S*_i_/*I*^2^ for several devices in both the HRS and LRS at 0.1 V. (**b**) The *S*_i_/*I*^2^ in both the HRS and LRS according to the RS cycle at the same read bias. (**c**) The change in the *S*_i_/*I*^2^ according to the RS cycle in both the HRS (left) and LRS (right) at the frequencies 20, 40, and 100 Hz.

## Data Availability

Not applicable.

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
