# Peer review of "Low-Frequency Noise-Based Mechanism Analysis of Endurance Degradation in Al/αTiO_x_/Al Resistive Random Access Memory Devices"

_materials, 2023, doi:10.3390/ma16062317_

Round 1
Reviewer 1 Report
See attached file

Reviewer 2 Report
1. How about the retention performance of the as-studied devices?
2. How many devices have been tested and how about the device stability?
3. The authors should do some other material characterizations, such as XPS and XRD.
Reviewer 3 Report
Manuscript ID: materials-2225351
Title- Low-Frequency Nosie-Based Mechanism Analysis of Endurance Degradation in Al/αTiOx/Al Resistive Random Access Memory Devices
The authors have studied the degradation of endurance using low frequency noise measurements in an Al/∝TiOx/Al RRAM device. The work is good but the following points need to be addressed before its publication:
1- English is poor. Authors are recommended to check for grammatical mistakes very carefully.
2- Under the introduction part, authors should add some more features of RRAM such as forming-free, intermediate resistance states and gradual RSEST process.
3- The authors should explain the reason for considering ∝TiOx as the active layer. It will be better to add some more lines about the selection of materials.
4- It is suggested to provide the results of the characterization of ∝ TiOx thin film such as XRD or Raman analysis under the results section.
5- In Al/∝TiOx/Al RRAM device, both top and bottom electrodes are same (i.e. Aluminum), so more discussion is needed as I-V curve is asymmetric.
6- The RESET and SET process is abrupt or gradual? It seems that both SET and RESET process are gradual in this device (as shown in fig. 2(a)). Authors should clear this type of switching. What is the resistance ratio (i.e. RHRS/RLRS) before and after degradation of the HRS.
7- There are some reported literature on resistive switching, so they need to point out the novelty in this work. They may consult the following papers.
https://doi.org/10.1016/j.mtcomm.2023.105484
https://doi.org/10.1016/j.physb.2023.414742
https://doi.org/10.1016/j.jpcs.2022.110689
8- Page No 1. , line no. 32, correct dynamic random access memory as dynamic random access memory (DRAM).
Page No. 2, Line No. 77, correct TEM as transmission electron microscopy (TEM).

Reviewer 4 Report
Overall, the manuscript is well written. The authors addressed an important yet interesting problem that is the major and critical shortcoming of interfacial-type RRAM, compared with the mainstream standalone 1T-1C DRAM modules. The write endurance degradation is a very important factor that limits the working lifetime of the entire RRAM die. The advancement of this work may help others design better wear leveling write mapping schemes for RRAM. Analyzing the device physical mechanisms of endurance degradation-induced write failure or stuck-at fault of memory cells during consecutive write operations is also very interesting. The corresponding RRAM device was fabricated and experimentally measured. One thing is that the achieved P/E cycling number is too low – only 500-times was achieved. It would be better to use heat-assisted acceleration to shorten the real device evaluation time of ordinary cycling numbers, e.g., 10^7. It’s good to see the manuscript publishing in the journal.
Round 2
Reviewer 1 Report
See attached file

Reviewer 3 Report
English may be improved a little more.
